# Carbon Emissions and Removals from Forests: New Estimates, 1990 2020

Francesco N. Tubiello[1], Giulia Conchedda[1], Nathan Wanner[1], Sandro Federici,[2] Simone Rossi[3] and Giacomo Grassi[3]

[1]Statistics Division, FAO, Rome, 00153, Italy
[2]Institute for Global Environmental Strategies, [3]IGES, Hayama, 240-0112, Japan
[3]European Commission Joint Research Centre, EC JRC, Ispra, 21027, Italy

*Correspondence*: Francesco N. Tubiello (francesco.tubiello@fao.org)

**Abstract.** National, regional and global $CO_2$ emissions and removals from forests were estimated for the period 1990–2020, using as input the country reports of the Global Forest Resources Assessment 2020. The new FAO estimates, based on a simple carbon stock change approach, update published information on net emissions and removals from forests in relation to: a) net forest conversion; and b) forest land. Results show a significant reduction in global emissions from net forest conversion over the study period, from a mean of 4.3 in the 1991–2000 to 2.9 Gt $CO_2$ yr$^{-1}$ in 2016–2020. At the same time, forest land was a significant carbon sink globally, but decreasing in strength over the study period, from -3.5 to -2.6 Gt $CO_2$ yr$^{-1}$. Combining net forest conversion with forest land, our estimates indicated that globally forests were a small net source of $CO_2$ to the atmosphere on average during 1990–2020, with mean net emissions of 0.4 Gt $CO_2$ yr$^{-1}$. The exception was the brief period 2011–2015, when forest land removals counterbalanced emissions from net forest conversion, resulting in a global net sink of -0.7 Gt $CO_2$ yr$^{-1}$. Importantly, the new estimates allow for the first time in the literature to characterize forest emissions and removals for the decade just concluded, 2011–2020, showing that in this period the net contribution of forests to the atmosphere was very small, i.e., a sink of less than -0.2 Gt $CO_2$ yr$^{-1}$— an estimate not yet reported in the literature This near-zero balance was nonetheless the result of large global fluxes of opposite sign, namely net forest conversion emissions of 3.1 Gt $CO_2$ yr$^{-1}$ counterbalanced by net removals on forest land of -3.3 Gt $CO_2$ yr$^{-1}$. Finally, we compared our estimates with data independently reported by countries to the United Nations Framework on Climate Change, indicating close agreement between FAO and country emissions and removals estimates. Data from this study are openly available via the Zenodo portal (Tubiello, 2020), with DOI https://10.5281/zenodo.3941973, as well as on the FAOSTAT Emissions database (FAO, 2021).

## 1 Introduction

Emissions from agriculture, forestry and other land uses represent nearly a quarter of world total anthropogenic emissions (Smith et al., 2014; IPCC 2019). Importantly, the $CO_2$ component of these emissions is generated on land at the margin between farm and natural ecosystems, largely in relation to processes that convert land for agricultural use, such as deforestation and

drainage of peatlands, generating roughly 4-5 Gt $CO_2$ yr$^{-1}$ in recent decades (e.g., Tubiello, 2019). Additional important anthropogenic emissions and removals of $CO_2$ are located directly on forest land, in relation to processes linked to forest management or degradation.

There is nonetheless significant disagreement between carbon cycle models on the one side, and national greenhouse gas inventories (NGHGI) on the other, on the quantification of the combined emissions and removals of $CO_2$ from all these land processes, though it is being increasingly shown that most differences are due to boundaries and definitional issues (e.g., Grassi et al., 2018; 2021). Greatly simplifying and limiting our scope to forests, terrestrial carbon cycle models have tended to focus on the $CO_2$ emissions from deforestation and forestry activities (land use change processes defined under the term $E_{LUC}$), while NGHGI have typically added removals on forest land beyond those linked to forestry practices, which the models tend not to consider anthropogenic. These forest removals in NGHGI counterbalance the positive emissions, resulting in near-zero estimated total net contributions of forests to the atmosphere (Grassi et al., 2018). Beyond the critical issues of the differences in boundaries and definitions between the two approaches, which are addressed elsewhere (e.g., Grassi et al., 2021), there is a significant need to improve the underlying activity input data used by both approaches. To this end, the Food and Agriculture Organization of the United Nations (FAO) collects, analyses and disseminates at regular intervals a wealth of country-based forest statistics through its Global Forest Resources Assessment (FRA), describing the status of forests with data at country, regional and global level (FAO, 2020a). FRA activity data of forest land area and carbon stock serve as critical inputs for estimates of forest carbon fluxes by FAO (Federici et al., 2015; FAO, 2020b) and other major international efforts (e.g., Friedlingstein et al., 2019; IPCC, 2019; Houghton and Nassikas, 2017). This paper describes the forest statistics available at FAO to estimate emissions and removals of $CO_2$ from forests that, being based on a simple though powerful (and replicable) carbon stock change method, generate data that can serve as boundary conditions to help evaluate more complex terrestrial carbon model results and NGHGI data. Our analysis highlights new trends based on the use of FRA 2020 input data, documenting the differences with respect to the previous use of FRA 2015. Finally, it compares results to national data independently reported by countries to the United Nations Framework Convention on Climate Change (UNFCCC).

## 2 Material and Methods

The estimates of $CO_2$ emissions and removals from forests made by FAO and published in FAOSTAT (FAO 2020b) are computed by applying a simplified carbon stock change method based on the 2006 IPCC Guidelines for National Greenhouse Gas Inventories (IPCC, 2006). Previous estimates covered the period 1990-2015, using as inputs activity data from the FRA 2015 (Federici et al., 2015). This work extends the FAO estimates of emissions and removals to 2020, by adding new input data for the period 2015-2020, while incorporating any revision in time series that may have occurred in the FRA 2020 with respect to FRA 2015. In describing the methods used in this work, we also discuss their limitations and uncertainties and the scope for comparing FAO estimates to UNFCCC country data.

**2.1 Gap-filling**

The FRA 2020 data used herein are: *forest land* area, as a total and for its two sub-categories, i.e., *Naturally regenerating forest* area (including both primary and secondary forest) and *Planted forest area*; and carbon stock in above and below-ground living biomass. Data cover the period 1990-2020.We gap-filled missing carbon stock data when needed, by using relevant regional averages of carbon stock density (carbon stock per unit forest land area), multiplied by country forest land area. Additionally, we checked the consistency of forest land area values against its two sub-components. In the few cases when such consistency was violated, we re-computed *naturally regenerating forest* area as the difference between *forest land* and *planted forest* area. The slightly revised dataset was used as input into the emissions calculations. It is openly available via the Zenodo portal (Tubiello, 2020), with DOI https://10.5281/zenodo.3941973, as well as via the FAOSTAT database (FAO, 2020a).

**2.2 Forest Definition**

The term *forest land* used herein follows the international FAO land use definitions (FAO, 2020b), also adopted by the UN system for environmental economic accounting (SEEA AFF, 2020), based on the FRA. As a land use category, the FAO definition of *forest land* comprises areas under forestry production, forest conservation including natural parks, and in general any area regulated administratively in terms of destination and use, including unmanaged forests, as long as three basic biophysical conditions are met, namely: i) minimum tree height of 5 m at maturity; ii) overall crown cover greater than 10%; and iii) minimum 0.5 ha in extension. (for complete definitions see, e.g., the FAO Land Use questionnaire, http://www.fao.org/economic/ess/ess-home/questionnaires/en/).

Countries reporting forest land data to FAO are expected to adhere to the above definitions and explicitly present how the conversion from national land use categories to the FAO categories was done. However, the uncertainty of the related area and stock estimates remains largely unreported. The magnitude of these uncertainties can vary significantly depending on the underlying estimation and/or mapping methodology. It has been recently shown that estimates based on land use and land cover information derived from remote sensing can results in differences of up to 20% at regional level, largely due to the difficulty of mapping land cover characteristics to land use status (FAO, 2020c). For well-defined forest land areas, typical uncertainties in national forest inventories may be nonetheless an order of magnitude smaller. For lack of additional knowledge of how uncertainty in local measurements carried out at national to regional levels, we applied the generic uncertainty suggested by IPCC for FAO activity data (20%) to the forest land area and biomass stock data used in this work.

In terms of comparison with UNFCCC data, we note that the FAO forest land use definitions used herein may differ from those used by countries for reporting their national GHG inventories (NGHGI), for instance in relation to minimum forest area thresholds or in criteria to assign land use status. Furthermore, country reporting to UNFCCC of emissions and removals data is limited to areas of managed forest, as per IPCC guidelines, while the FAO land use definitions comprise both managed and unmanaged forests, as discussed above. In practice, such differences may often be small, considering that a large portion of

the world's forest land area in many countries is administratively regulated. Finally, we note that the FAO forest land area
considered herein does not track separately, as done instead in UNFCCC reporting, the two- sub-components *forest land*
*remaining forest land* (FL-FL) and newly converted forest land. This is often overlooked in the literature, where FAO estimates
of forest land emissions and removals may be incorrectly compared to UNFCCC data for FL-FL (e.g., Petrescu et al., 2020).
**2.3 Emissions and Removals**
The estimates presented herein provide information on total net emissions and removals from forests, in total as well as by
component processes. Specifically, for each country $a$ and total carbon stock $B_a$, the total forest emissions/removals, $ER_a$, were
computed as a simple carbon stock change, as follows:
$$ER_a(t_i) = - \Delta C_a(t_i) = - [ B_a(t_i) - B_a(t_{i-1})] = NFC_a(t_i) + FL_a(t_i) \tag{1}$$
Where biomass stock information was derived from the FRA 2020 as indicated in the previous section, and $t_i$ = 1990, 2000,
2010, 2015, 2020 represent FRA years. The minus sign was used to adhere to the convention of considering emissions as
positive fluxes to the atmosphere, corresponding to decreases in forest carbon stock— and vice-versa to consider removals as
negative fluxes, i.e., from the atmosphere into forest land, corresponding to increases in forest carbon stock. We note that the
estimates in equation (1) are robust as well as easily replicable by anyone having access to FRA data. At the same time, it is
noted that the FAO carbon stock change estimates include only two of the five carbon pools typically reported by countries
according to IPCC. This difference may affect the magnitude of the estimated C stock changes, although likely not the sign,
because of biophysical linkages across carbon pools. The net forest signal to the atmosphere, ER, was split into two mutually
exclusive components, specifically emissions from *net forest conversion*, NFC, and emissions/removals from *forest land*, FL
(Fig. 1).
**2.3.1 *Emissions from Net Forest Conversion***
For each country $a$, total carbon stock $B_a$, and time period $t_i$, the emissions from *net forest conversion*, $NFC_a(t_i)$ in equation (1)
were computed as the positive carbon flux to the atmosphere associated with net forest land area loss, tracked separately for
sub-categories *naturally regenerating forest*, $NR_a$, and planted forest, $PL_a$ as follows:

$$NFC_a(t_i) = - [ B_a(t_{i-1})/A_a(t_{i-1}) ] *\{Min [NR_a(t_i) - NR_a(t_{i-1}) , 0 ] + Min [PL_a(t_i) - PL_a(t_{i-1}) )] , 0] \} \tag{2}$$

Thus net forest conversion tracks losses of both naturally regenerating (including primary and secondary forests) and planted
forest areas. It should be noted that in cases when net forest land area change is positive, indicating net area gains, NFC is zero
by definition and the relevant emissions/removals are instead accounted for on forest land (see next section). A number of

limitations apply to the computation of emissions in (2), First, results are limited by the lack of carbon stock data by forest sub-component, resulting in the need to apply a single value for both naturally regenerating forest and planted forest. Considering that the majority of forest area losses in the FRA 2020 pertain to the natural forest component, however, the use of a single carbon density value in (2) is not a significant issue to this end. At the same time, carbon stock density can be expected to be higher in natural forests than the average biomass stock (which also includes carbon stock in plantations), implying that the NFC emissions computed in (2) are likely underestimates. Furthermore, we note that equation (1) above does not depend on the availability of carbon stock values by forest sub-component. A second important limitation to equation (2) is that forest losses are computed net of forest area gains taking place over the same period. The underlying activity data used as input do not in fact allow separate tracking of gross gains and losses. Thus in terms of comparison to UNFCCC, FAO *net forest conversion* data would roughly correspond to the sum of UNFCCC-reported land use changes from forest land to non-forest land, for those countries using the so-called 'IPCC approach 1' to land use representation, which like our estimates relies on net area changes. By contrast, use of more accurate national forest inventories, with more detailed identification of gross area fluxes, would generate larger differences between FAO estimates and the corresponding UNFCCC country data for this category. Finally and importantly, estimates in equation (2) are limited by the underlying uncertainty in the activity data. Simple error propagation of the component uncertainties in area and carbon stock discussed in the previous section give an uncertainty in NRC emissions of roughly 50%. This is consistent with values used for land use change emissions estimates published in recent IPCC reports (IPCC, 2019) and in relevant carbon cycle literature (Friedlingstein et al., 2019). For coherence, we applied this uncertainty value to ER and FL estimates.

### 2.3.2 *Emissions and Removals on Forest Land*

For any country $a$, total carbon stock $B_a$, and time period $t_i$ in equations (1) and (2) above, the emissions/removals on *forest land*, $FL_a(t_i)$, were computed as the residual between total forest carbon stock change and net forest conversion, as follows:

$$FL_a(t_i) = ER_a(t_i) - NFC_a(t_i) \qquad (3)$$

The emissions/removals computed in (3) represent the net carbon flux to or from the atmosphere located within the boundaries of forest land area, arising from a combination of carbon stock and forest area changes between successive FRA periods. These changes in principle may arise from both anthropogenic and natural causes, including legacy effects of deforestation prior to the study period, afforestation, forest management, climate signals, as well as the impacts on plant growth of nitrogen deposition and increased atmospheric $CO_2$ concentrations. As discussed above, we associated an uncertainty level of 50% to estimates in equation (3), consistently with those computed for the emissions from net forest conversion and in line with the uncertainty used in the literature.

Within the differences highlighted above, with regards to land accounting approaches and differences in national forest
definitions, the FAO emissions/removals on *forest land* largely correspond to those used by countries in their reporting to
UNFCCC with respect to forest land.
**2.4 Comparisons to UNFCCC data**
A final consideration on the limitations of the approach presented herein concerns the underlying drivers of the
emissions/removals estimates, i.e., whether they could be labelled as anthropogenic or natural fluxes. On the one hand, the
definitions underlying equation (1)-(3) make the association impossible within our approach. On the other, a bit more can be
said in practice. This is because human intervention is typically required to determine land use changes—for instance the
establishment of specific activities, for instance agriculture, preventing natural forest regrowth and recovery following forest
loss. To this end, and within the limitations discussed above, *net forest conversion,* representing permanent forest loss in the
FAO statistics, can be considered virtually all anthropogenic in nature, hence a good proxy for human-driven deforestation.
Conversely, only a portion of the emissions/removals estimated on forest land can be considered anthropogenic. At the same
time, recent work shows that the anthropogenic portion of this component can be substantial, once the concept of 'managed
land' is expanded beyond forestry practices to include all forest areas except in very remote places (Grassi et al., 2021).
Nonetheless, because of the above complexities, we chose not to determine *a priori* the anthropogenic portion of our
emissions/removals estimates. Instead, we complemented our analysis of results with a comparison between our estimates of
emissions and removals and the anthropogenic fluxes submitted by countries to UNFCCC. In this context, although it is
recognized that countries report data to both FAO and UNFCCC, we reserve herein the term 'country data' to the
emissions/removals reported by countries to the UNFCCC.
To this end, we used country data accessed at the UNFCCC data portal (UNFCCC, 2020) and complemented with information
from national Biennial Update Reports (BURs). While data from Annex I countries (AI) are fairly complete over the period
1990– 2018, data from non-Annex I (NAI) countries are sparse, although becoming increasingly available through BURs.
Given these data limitations, a full comparison was possible only for Annex I countries for the FRA periods 1990–2000; 2001–
2010; and 2011– 2015. First, we compared results of equation (3) with aggregate Annex I reporting of emission/removal for
the category '4.A Forest land' (UNFCCC, 2020). To gain further insights, we also separately analyzed emissions/removals on
forest land reported by individual countries in their national GHG inventories (NGHGIs), focusing on those reporting large
sinks, i.e., Canada, Russian Federation and the United States of America among Annex I parties, and China among non-Annex
I parties. We also compared our results for net forest conversion to available non-Annex I country data from Brazil and
Indonesia, representing large emission sources, according to FAOSTAT estimates respectively the first and third emitters in
this category (FAO, 2020b). Unfortunately, no BUR submissions have been made so far by the Democratic Republic of
Congo—the second largest emitter from deforestation according to FAOSTAT data—which therefore could not be included
in this comparison exercise. Data for NAI countries were sourced from China's second Biennial Update Report (2018), Brazil's
third Biennial Update Report (2019) and from Indonesia's second Biennial Update Report (2018).
**2.5 Structure of the datasets on emissions-forest land and online access**
The FAO emissions and removals estimates and associated area information statistics are disseminated in the FAOSTAT
Emissions Land Use/ Forest Land domain as yearly statistics, over the period 1990–2020 (FAO, 2021), for 220 countries and
territories. Annual mean fluxes are obtained by dividing the outcomes of (1)-(3) by the relevant time-period underlying FRA
intervals, i.e., by 5 or 10 years. They therefore refer to the following periods: 1991–2000; 2001–2010; 2011–2015; and 2016–
2020. For completeness, values for the year 1990 were set equal to the averages computed for 1991–2000, and the full period
of analysis was referred to as 1990-2020. Data include, by country and year, forest land area and area of net forest conversion
(in 1000 ha), emissions from net forest conversion; emissions/removals on forest land; and total emissions/removals from
forests (in kt $CO_2$). The carbon stock in living biomass (in Mt C) is available under the FAOSTAT database, Inputs/Land Use
(FAO, 2020c). Data are disseminated by country, by standard FAO regional aggregations and special groups, including the
Annex I and non-Annex I country grouping relevant to UNFCCC reporting.
**3 Results**
We present below the main findings of annual $CO_2$ emissions/removals estimates from net forest conversion, forest land, and
their aggregate, total emissions and removals from forest, for the period 1990–2020, computed for more than 200 countries
and territories, based on equations (1)-(3) above. Emissions and removals are expressed in annual means (Gt $CO_2$ yr$^{-1}$) relative
to the relevant FRA period. Results are presented at global level, by Annex I and non-Annex I countries and by region, where
relevant. Differences with estimates based on earlier FRA 2015 input data are also discussed, where of interest.
**3.1 Emissions from Net Forest Conversion**
Results show that carbon fluxes to the atmosphere from *net forest conversion* were significant, with world-total means of 3.7
Gt $CO_2$ yr$^{-1}$ for the period 1990—2020, and almost entirely located in non-Annex I countries, which contributed more than 90
% of the world total (Table 1). In terms of temporal trends, the global mean decreased by 20% from 1990 to 2015, from 4.3 to
3.3 Gt $CO_2$ yr$^{-1}$, less than previously estimated over the same period using the FRA 2015 (-40 %). It decreased by another 10%
to 2.9 Gt $CO_2$ yr$^{-1}$ during 2016–2020. For the period 2016–2020, the Americas and Africa were nearly equal contributors, but
with markedly opposite trends compared to the period 1991–2000. Specifically with respect to the two time periods, emission
in the Americas nearly halved, from 2.2 to 1.3 Gt $CO_2$ yr$^{-1}$, while they increased in Africa, from 0.9 Gt to 1.1 $CO_2$ yr$^{-1}$. Asia
was the third region in terms of emissions from net forest conversion, showing a slight decrease, from 0.6 Gt to 0.4 $CO_2$ yr$^{-1}$
over the same time periods (Fig. 2).
**3.2 Emissions and removals on forest land**
Emissions/removals on forest land showed a net sink over the entire period 1990–2020, with a mean removal of -3.3 Gt $CO_2$
yr$^{-1}$ globally. This forest carbon flux was nearly equally divided between Annex I (-1.8 Gt $CO_2$ yr$^{-1}$) and non-Annex I countries
(-1.5 Gt $CO_2$ yr$^{-1}$) (Table 1). Additionally, we computed that the new FAO estimates indicated a stronger forest sink than
previously estimated using FRA 2015 data, i.e., on average 1.0 Gt $CO_2$ yr$^{-1}$ (35 %) stronger, due to larger estimated sinks in
Europe (dominated by trends in Russian Federation) and Asia (China).
At the same time, the estimated global forest land sink weakened in strength over the study period, with the world total mean
decreasing from -3.5to -2.6 Gt $CO_2$ yr$^{-1}$, i.e., about 20 % decrease from 1990 to 2020. The period 2011–2015 represented an
exception to this decreasing trend, showing the strongest forest sink over the study period, with mean world total rates of -4.0
Gt $CO_2$ yr$^{-1}$. In terms of regional distribution and averaged over the period 1990–2020, Europe, the Americas and Asia nearly
equally contributed to the estimated forest land removals, within a narrow range of -1.0 to -1.2 Gt $CO_2$ yr$^{-1}$, with Europe
(including the Russian Federation) being the largest contributor. Conversely, forest land in Africa was a source to the
atmosphere, with mean emissions increasing significantly from 2000 to 2015, i.e., from 1.4 to 43 Mt $CO_2$ yr$^{-1}$ (Fig. 3). By
associating net forest land emission to forest degradation, as done in Federici et al. (2015), our results suggest a significant
relative increase in forest degradation (defined as carbon stock reduction in forest land) in Africa over the last twenty years.
**3.3 Total emissions and removals from forests**
Our estimates show that the net effects of emissions from net forest conversion and removals on forest land were a small net
source of $CO_2$ emissions to the atmosphere, with a world total mean of 0.4 Gt $CO_2$ yr$^{-1}$ over the 1990–2020 period. This new
estimated value was significantly less than reported earlier based on FRA 2015 data (Table 1). It is further of interest to note
that the estimated small global source was the result of a balance of larger fluxes: a net sink on forest land, largely located in
in UNFCCC Annex I countries (-1.5 Gt $CO_2$ yr$^{-1}$), counterbalanced by a net emission source from net forest conversion, mainly
in non-Annex I countries (1.9 Gt $CO_2$ yr$^{-1}$).
A more detailed analysis focusing on trends over time (Fig. 4) revealed two notable new findings of our analysis with respect
to previous results. First, the period 2015-2020 saw a reversal of the decreasing trend in non-Annex I sources and the increasing
trend in Annex I sinks seen for the period 1990 to 2015. Specifically, non-Annex I sources from net forest conversion began
to increase again in 2016-2020, from 1.3 to 1.6 Gt $CO_2$ yr$^{-1}$, while Annex I sinks on forest land began decreasing in strength,
from -2.0 to -1.3 Gt $CO_2$ yr$^{-1}$. Second, and remarkably, forests acted as a net overall sink of atmospheric $CO_2$ during 2011–
2015, averaging nearly -0.7 Gt $CO_2$ yr$^{-1}$, largely a result of decreased emissions from net forest conversion in this period,
counterbalanced by a strong sink on forest land. Conversely, FAO had previously estimated for the same period, based on FRA
2015 input data, a net emission source of 1.1 Gt $CO_2$ yr$^{-1}$ (Table 1).
**3.4 Comparisons with UNFCCC**
*Forest Land*
As discussed in the methodology section, we first compared our estimates of emissions/removals on forest land to data reported
by Annex I countries, i.e., for category "4.A Forest land" in their national GHG inventory (UNFCCC, 2020). In the aggregate,
e.g., summing up all country data and averaging over the period 1990–2015, our estimates agreed in both sign and magnitude
with the UNFCCC country data (14 % relative absolute error). Specifically, our estimates indicated a mean sink of -1.8Gt $CO_2$
yr$^{-1}$ vs -2.2 Gt $CO_2$ yr$^{-1}$ reported. Using the FRA 2015 in earlier work (Federici et al., 2015) had given a 33 % smaller sink
(Table 2). Our estimates were particularly well aligned with country reporting for the period 2010–2015, i.e., within 5 %,
predicting a sink on forest land of -2.1 Gt $CO_2$ yr$^{-1}$ vs -2.2 Gt $CO_2$ yr$^{-1}$ reported. As in the previous case, earlier sink estimates
based on the FRA 2015 were 40 % smaller (Fig. 5).
Comparisons of estimated emissions/removals on forest land for specific countries with large reported sinks confirmed the
good overall agreement found for Annex I parties in aggregate. For instance, on average over the period 1990–2015, our
estimates of forest land sinks for the Russian Federation were within 5 % of those reported by the country NGHGI. Agreement
with NGHGI data was even closer after the year 2000, i.e., for the period 2001–2010 our estimates indicated a mean sink on
forest land of -800 Mt $CO_2$ yr$^{-1}$ versus country data of -750 Mt $CO_2$ yr$^{-1}$, and a mean sink of -730 Mt $CO_2$ yr$^{-1}$ versus -680 Mt
$CO_2$ yr$^{-1}$ for the period 2011–2015 (Fig. 6). Comparisons for the USA were also encouraging, albeit with larger differences
than found for the Russian Federation. On average over the period 1991–2010, the FAO estimates were of a 25 % smaller sink
on forest land compared to the NGHGI country data. Averaged over the period 2011–2015 our estimates were 29 % smaller
than the country data, or -460 Mt $CO_2$ yr$^{-1}$ and -650 Mt $CO_2$ yr$^{-1}$, respectively.
We performed comparisons for China, using data from the country's Second Biennial Update Report (2018), to extend our
analysis to non-Annex I countries reporting large sinks on forest land. Specifically, we used national data on total removals
from LULUCF for the period 2011–2015. We concluded that China LULUCF data were a good proxy for forest land data,
considering that: 1) zero emissions from net forest conversion were indicated in the same BUR; and 2) emissions from cropland
and grassland—the other main component of LULUCF within a national inventory—were likely small, as indicated by
independent emissions estimates published in FAOSTAT (FAO, 2020b). Within these assumptions, our estimates of a sink on
forest land in China for the period 2011–2015 agreed well with country data (within 20 % of country data), i.e., -710 Mt $CO_2$
yr$^{-1}$ compared to -840 Mt $CO_2$ yr$^{-1}$ reported to UNFCCC (Fig. 6).
Conversely, our estimates of emissions/removals on forest land did not agree well to those reported by Canada. Our results
indicated a net source on forest land, declining from 2000 to 2015, whereas the NGHGI country data reported a progressively
smaller sink over the same period (Fig. 6). Specifically for the period 2011–2015, our estimates indicated a weak net source,

about 23 Mt $CO_2$ yr$^{-1}$, compared to a net sink of -150 Mt $CO_2$ yr$^{-1}$ in the country data. Finally, our estimates for the most recent period, 2016–2020, for which however there is no available NGHGHI data yet from the country, began to show a sink on forest land, of -80 Mt $CO_2$ yr$^{-1}$, thus indicating a possible alignment with NGHGI data in recent years. A possible reason for the discrepancies found in this case may relate to differences in land use definitions, particularly those related to managed forest land. For the purpose of the NGHGI, in fact, the area of managed forests defined by Canada is 65 % of the total forest land area reported to FAO (Canada's 7th National Communication and 3rd Biennial Report, 2017; Ogle et al., 2018).

*Net forest conversion*

We also compared estimates of emissions from net forest conversion with data reported to UNFCCC. As discussed in the methodology section, FAO estimates of emissions from net forest conversion are proxies for deforestation emissions data. The two countries for which relevant data were available were Brazil and Indonesia. For Brazil, we compared our estimates of net forest conversion directly to deforestation emissions from the country's BUR. For Indonesia, we compared our estimates to sum of LULUCF emissions arising from land use change to cropland and grassland—assuming, in line with current understanding of deforestation trends in this country, that land converted to cropland and grassland in Indonesia originated largely from loss of forest land area. For Indonesia, for the period 1991–2000, our estimates of emissions from net forest conversion greatly overestimated country data for deforestation, by over a factor of 10 (Fig. 7). Conversely, for the more recent period 2011–2015, they were on average within 25 % of country data, specifically 180 Mt $CO_2$ yr$^{-1}$ vs country data of 165 Mt $CO_2$ yr$^{-1}$. Our estimates further suggested a 50% decrease in emissions from net forest conversion in the period 2016-2020, for which however BUR data are not yet available (Fig. 7).

For Brazil, our estimates were in good agreement (within 10 %) of country data over the period 1990 to 2015, i.e., on average 1.4 vs. 1.5 Mt $CO_2$ yr$^{-1}$ reported data (Fig. 7). More in detail by decade, our estimates were 1.4 vs 1.9 Gt $CO_2$ yr$^{-1}$ during 1991–2000 and 1.6 vs 1.6 Gt $CO_2$ yr$^{-1}$ over 2001–2010. Conversely, for the period 2010–2015, our estimates of emissions from net forest conversion were significantly higher than reported in the BUR.

## 4. Discussion

The availability of new forest area and carbon stock data from the FRA 2020 enabled a new analysis of the role of forests in generating $CO_2$ emissions and removals at country, regional and global level, during the period 1990–2020. In particular, the new information allowed us, for the first time in the literature, to estimate emissions and removals relative to the most recent decade, covering the period 2011–2020. Our findings indicate that in the decade just concluded the net contribution of forests to the atmosphere, representing the combination of emissions from net forest conversion and removals on forest land, was very small, i.e., an overall emission sink of less than -0.2 Gt $CO_2$ yr$^{-1}$, estimated for the first time in the literature for this period. It nonetheless resulted from the balance of large global fluxes of opposite sign, namely mean net forest conversion emissions of 3.1 Gt $CO_2$ yr$^{-1}$, counterbalanced by mean net removals on forest land of -3.3 Gt $CO_2$ yr$^{-1}$ (Table 1). Both fluxes, and hence

the overall net near zero balance for forests, were shown to be in very good agreement with the data reported by countries in national GHG inventories, and in line with independent findings by Grassi et al. (2021). At the same time, the consistency of our estimates with those of terrestrial carbon cycle models were limited to the anthropogenic carbon flux from forests to the atmosphere (i.e., IPCC, 2019). Results further showed that, with respect to the previous decade 2001–2010, emissions from net forest conversions had decreased by 15 %, while removals on forest land had decreased by 5 %. Further analysis of the underlying FRA 2020 data (not shown) indicated that such decreases were due to a reduced pace of natural expansion and afforestation in Annex I countries, which have functioned historically (1990-2020) as forest sinks, as well as a decrease in forest loss in non-Annex I countries, which have represented the bulk of deforestation. The new estimates also show that over the earlier period 1991–2010 forests were a smaller net source of emissions than previously calculated (Federici et al. 2015). largely due to much stronger sinks on forest land estimated using the new FRA 2020 as opposed to FRA 2015 data, respectively for Europe (+ 0.7 Gt $CO_2$ $yr^{-1}$) and Asia (+ 0.6 Gt $CO_2$ $yr^{-1}$).

The main new finding of this work is the large estimated sink on forest land over the period 2011–2015, averaging -4.0 Gt $CO_2$ $yr^{-1}$, causing the overall net negative carbon flux from forests highlighted in the results section. Notable contributors to this forest land sink were the Russian Federation, USA, China, Indonesia and India, which all had stronger carbon uptake compared to the previous 2001–2010 period. Comparisons with country data reported to the UNFCCC support our estimates, indicating that they represent an improvement compared to previous results. In particular, the good agreement between our new estimates and country NGHGI data on emissions/removals on forest land and emissions from net forest conversion suggests that the definition of forest land area underlying both FAO and UNFCCC reporting was consistent across the countries considered, i.e., they considered most of the forest land area reported to FAO as managed for UNFCCC purposes—confirming the analysis provided in the methodological section of this paper. This implies that, limited to the countries tested and within the range of limitations discussed earlier in this paper, the estimates of emissions and removals from forests provided in this paper can be considered largely anthropogenic. Finally, the good agreements found between our estimates and country reports support the finding of a large anthropogenic sink on forest land for the period 2011–2015, leaving open the possibility, in need of verification in coming years, that even when considering deforestation, the world forests were a small sink, rather than a source of atmospheric carbon during this period. In fact, the discussed progressive reduction of the overall forest source observed across the two most recent decades is consistent with these findings.

## 5. Data availability

The emissions and removals data, alongside with input activity data of forest land area and carbon stock, are disseminated in FAOSTAT (FAO, 2021). An exact replica of the data used for this paper is available as open access at http://doi.org/10.5281/zenodo.3941973 (Tubiello, 2020).

## 6. Conclusions

Estimates of $CO_2$ emissions and removals from forests were updated based on the most recent FRA 2020 data and by applying a simple yet robust, transparent and easily replicable carbon stock change approach. Over the period 1990–2020, result confirmed known country, regional and global trends, providing additional detail to specific dynamics while extending available information to the period 2016–2020. Importantly, the new estimates allowed to characterize for the first time forest emissions and removals for the decade just concluded, 2011-2020, showing that in this period the net contribution of forests to the atmosphere was very small, sink i.e., less than -0.2 Gt $CO_2$ yr$^{-1}$. This near-zero balance was nonetheless the result of large global fluxes of opposite sign, namely net forest conversion emissions of 3.1 Gt $CO_2$ yr$^{-1}$ counterbalanced by net removals on forest land of -3.3 Gt $CO_2$ yr$^{-1}$.

**Author contributions.**

**Competing interests.** The authors declare that they do have no conflict of interest.

**Disclaimer.** The views expressed in this publication are those of the authors and do not necessarily reflect the views or policies of FAO.

**Acknowledgments.** This work was made possible by regular funding provided to FAO by its member countries. We thank our colleagues from the FAO Forestry Division who provided data, insight and expertise that greatly assisted our research. We are grateful to staff of the FAO Statistics Division for overall support, and in particular Giorgia De Santis for her preliminary analysis of the FRA 2020 data used as input into this work, Griffiths Obli-Layrea for UNFCCC data provision and Amanda Gordon for FAOSTAT data maintenance and dissemination. We are likewise thankful to the Forest Resources Assessment Process and its funders, for provision of data that made this work possible. The views expressed in this publication are those of the authors and do not necessarily reflect the views or policies of FAO.

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

| | FRA 2020 | | | FRA 2015 | | |
|---|---|---|---|---|---|---|
| | ER | NFC | FL | ER | NFC | FL |
| **1991—2000** | **0.8** | **4.3** | **-3.5** | **1.8** | **4.7** | **-2.9** |
| Annex I countries | -1.4 | 0.3 | -1.7 | -1.0 | 0.2 | -1.2 |
| Non-Annex I countries | 2.2 | 3.9 | -1.7 | 2.8 | 4.5 | -1.7 |
| **2001—2010** | **0.5** | **3.7** | **-3.1** | **1.2** | **3.7** | **-2.6** |
| Annex I countries | -1.6 | 0.3 | -1.9 | -1.4 | 0.4 | -1.8 |
| Non-Annex I countries | 2.1 | 3.4 | -1.3 | 2.6 | 3.3 | -0.8 |
| **2011—2015** | **-0.7** | **3.3** | **-4.0** | **1.1** | **2.9** | **-1.9** |
| Annex I countries | -2.0 | 0.2 | -2.1 | -1.1 | 0.1 | -1.3 |
| Non-Annex I countries | 1.3 | 3.1 | -1.8 | 2.2 | 2.8 | -0.6 |
| **2016—2020** | **0.3** | **2.9** | **-2.6** | | | |
| Annex I countries | -1.3 | 0.2 | -1.6 | | | |
| Non-Annex I countries | 1.6 | 2.7 | -1.1 | | | |
| **2011—2020** | **-0.2** | **3.1** | **-3.3** | | | |
| Annex I countries | -1.7 | 0.2 | -1.9 | | | |
| Non-Annex I countries | 1.5 | 2.9 | -1.5 | | | |
| **AVERAGE 1990—2020** | **0.4** | **3.7** | **-3.3** | | | |
| Annex I countries | -1.5 | 0.3 | -1.8 | | | |
| Non-Annex I countries | 1.9 | 3.4 | -1.5 | | | |
| | | | | | | |
| **AVERAGE 1990—2015** | **0.4** | **3.8** | **-3.4** | **1.4** | **4.0** | **-2.5** |
| Annex I countries | -1.6 | 0.3 | -1.8 | -1.2 | 0.3 | -1.4 |
| Non-Annex I countries | 2.0 | 3.6 | -1.6 | 2.6 | 3.7 | -1.1 |

4    **Table 2.** Estimates of emissions/removals on forest land for Annex I countries, based on FRA 2020 and FRA 2015, compared to country data

5    reported to UNFCCC (Gt $CO_2$ yr$^{-1}$).

| Annex I total emissions/removals | | | |
|---|---|---|---|
| | **FRA 2020** | **FRA 2015** | **UNFCCC** |
| 1991-2000 | -1.7 | -1.2 | -2.1 |
| 2001-2010 | -1.9 | -1.8 | -2.1 |
| 2011-2015 | -2.1 | -1.3 | -2.2 |
| 2016-2020 | -1.6 | | |
| **AVERAGE 1991—2015** | **-1.8** | **-1.4** | **-2.2** |

1 **Figures**

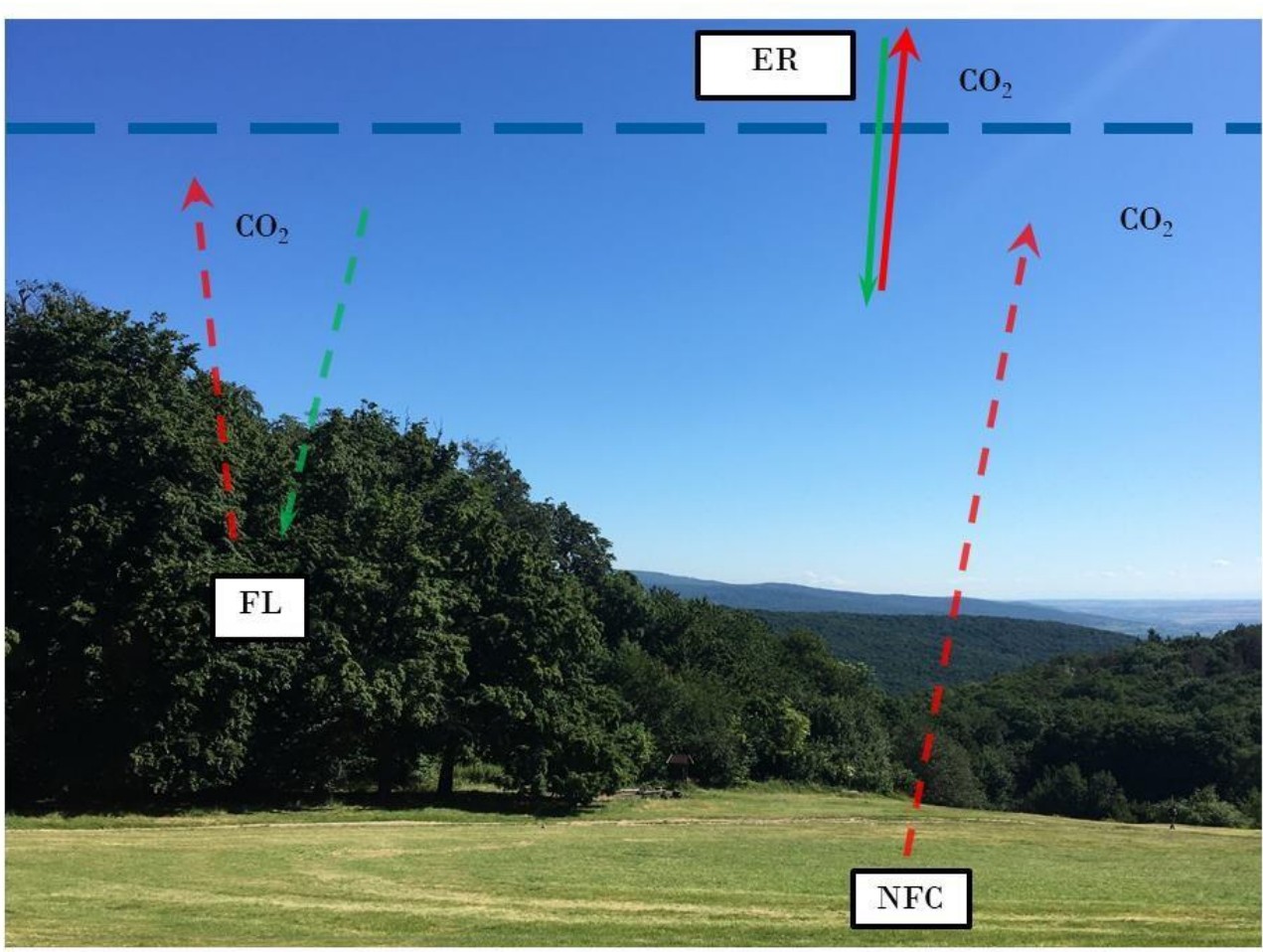

**Figure 1. The three main carbon fluxes considered in this paper, consisting of emissions from net forest conversion (NFC), emissions and removals on forest land (FL) and their aggregate, representing total net emissions/removals from forests (ER). Photo copyright: Francesco N. Tubiello.**


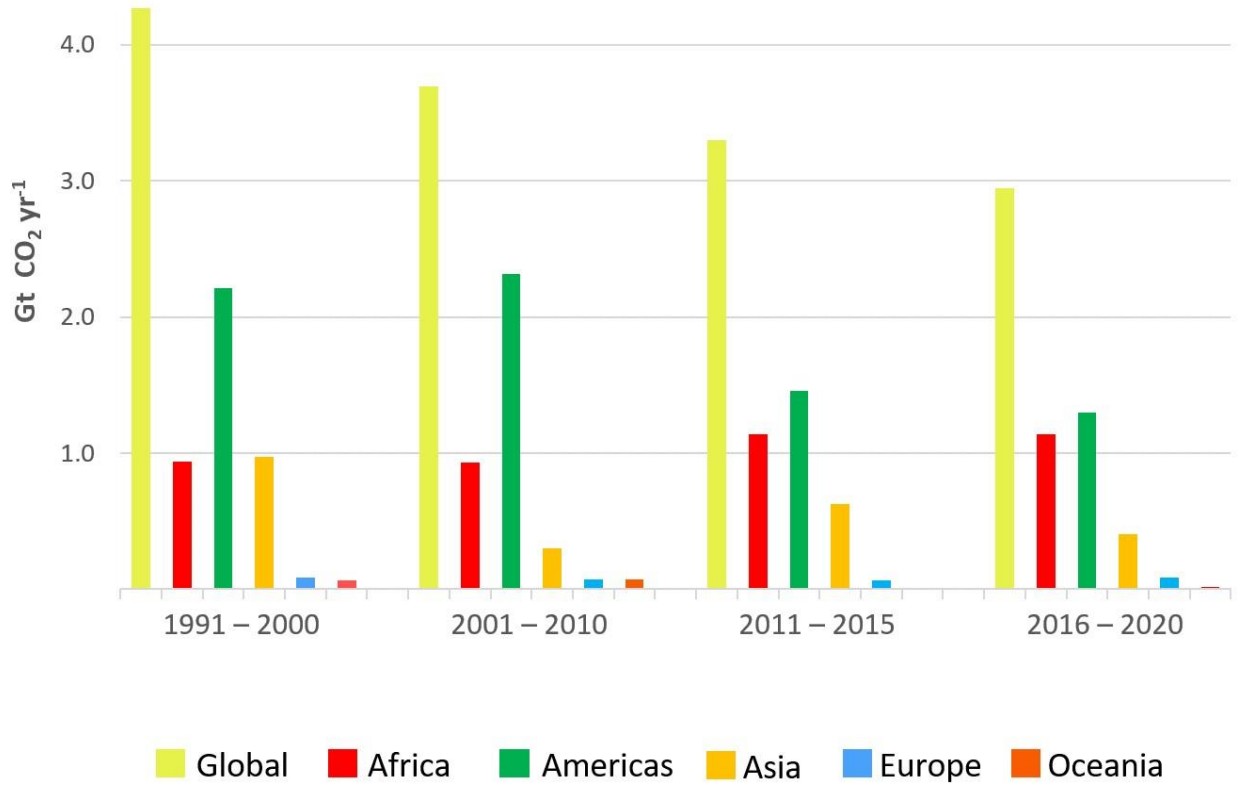

**Figure 2. Estimates of emissions from net forest conversion (NFC) based on FRA 2020 for global (acid green) and regional (Africa = red; Americas = green; Asia = gold; Europe = sapphire; Oceania = orange) totals, in Gt CO$_2$ yr$^{-1}$.**

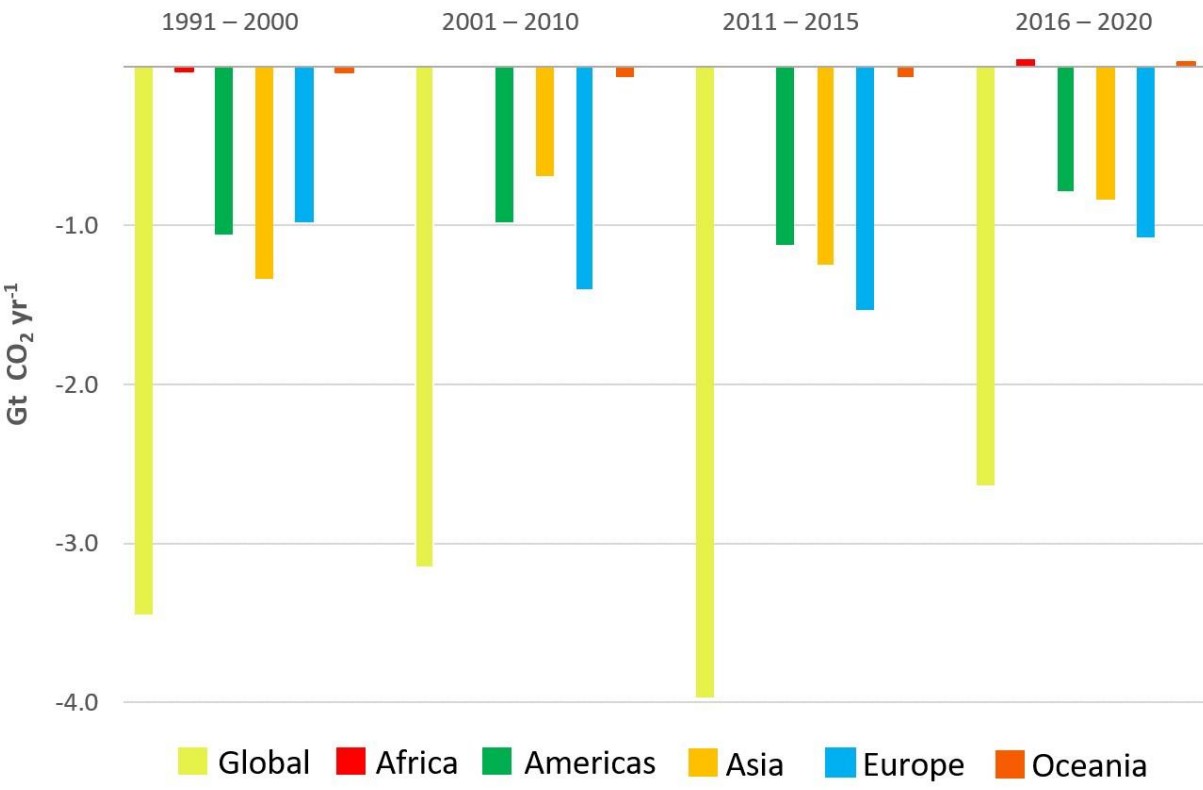

**Figure 3. Estimates of the emissions/removals on forest land (FL) based on FRA 2020 for global (acid green) and regional**

**totals (Africa = red; Americas = green; Asia = gold; Europe = sapphire; Oceania = orange), in Gt $CO_2$ yr[-1].**

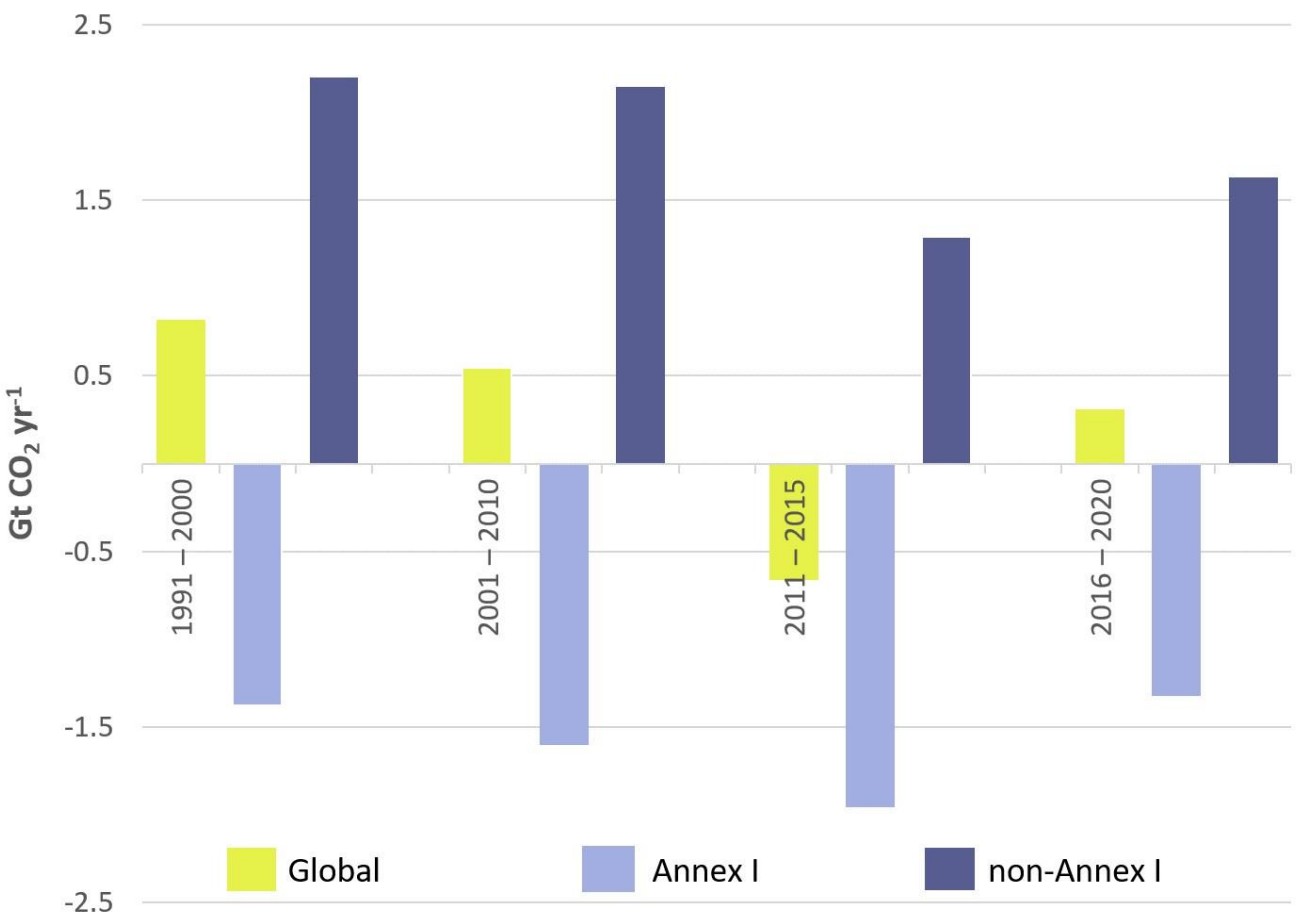

**Figure 4. Estimates of total emissions/removals from forests (ER), based on FRA 2020, for global (acid green), Annex I (lavender) and non-Annex I (purple navy) totals, in Gt $CO_2$ yr$^{-1}$.**

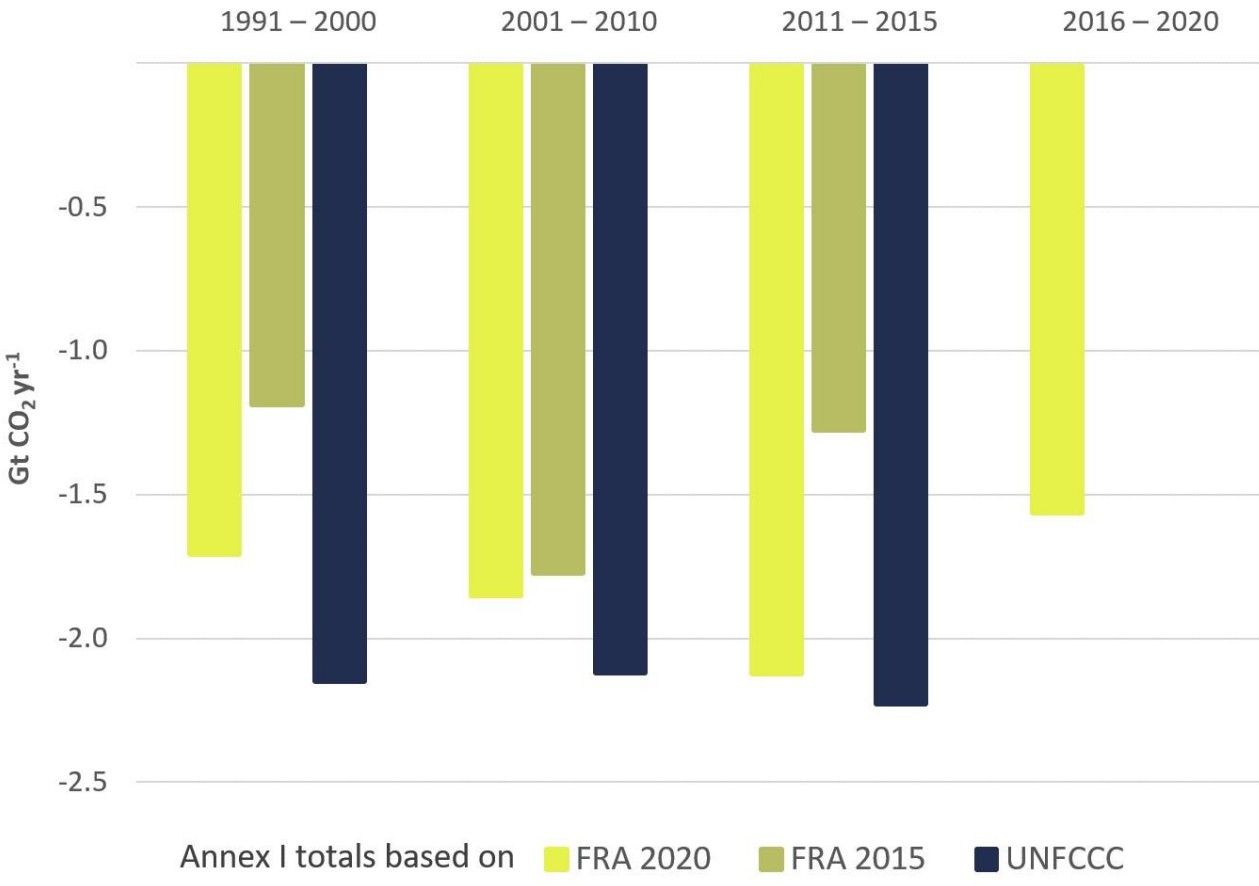

**Figure 5. Comparison of estimates of emissions/removals on forest land (FL) for Annex I totals, in Gt $CO_2$ $yr^{-1}$, based on FRA 2020 (acid green) and FRA 2015 (olive green), to the Annex I totals reported by countries to UNFCCC (cadet blue).**

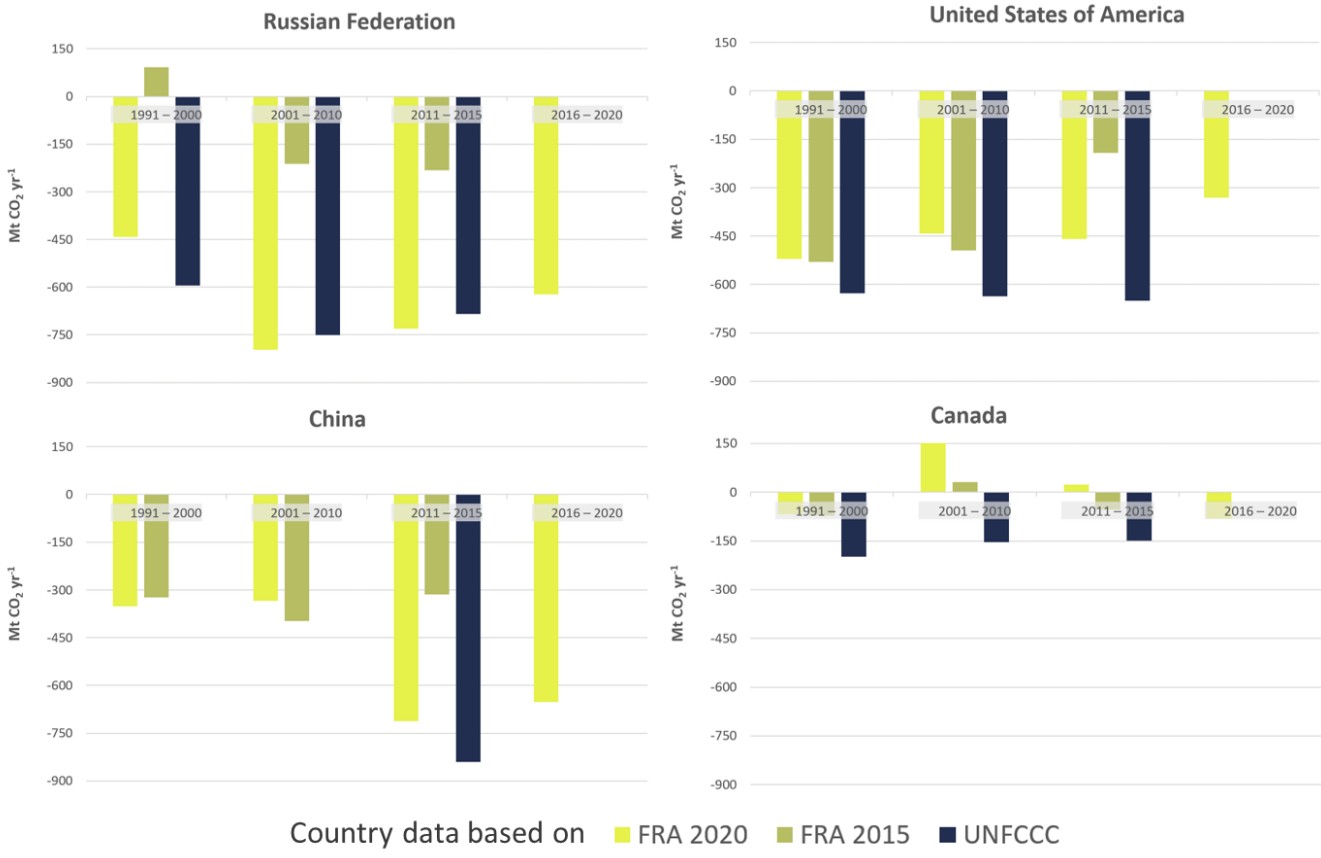


**Figure 6. Comparison of estimates of emissions/removals on forest land (FL) for Russian Federation (top left), USA (top right), China (bottom left) and Canada (bottom right), in Mt CO₂ yr⁻¹, based on FRA 2020 (acid green) and FRA 2015 (olive green), to country data reported to UNFCCC (cadet blue).**

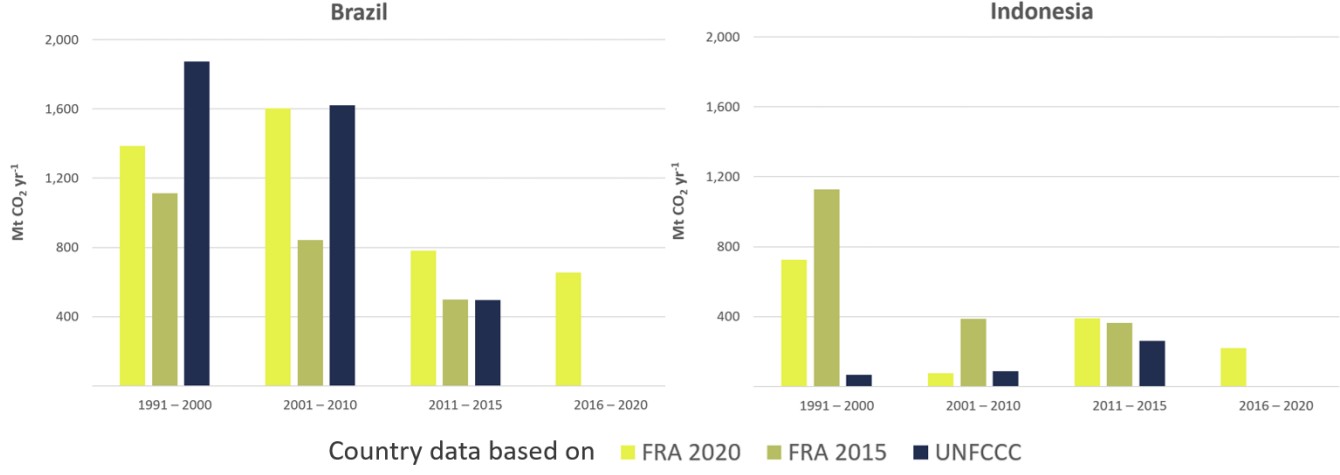

**Figure 7. Comparison of estimates of emissions from net forest conversion (NFC) for Brazil (left) and Indonesia (right), in**
**Mt CO$_2$ yr$^{-1}$, based on FRA 2020 (acid green) and FRA 2015 (olive green), to  country data reported to UNFCCC for**
**deforestation (cadet blue).**

