# Peer review of "Carbon Emissions and Removals by Forests: New Estimates 1990-2020"

_Earth System Science Data, 2020_

## Short Comment (SC1) · 14 Oct 2020

A general comment on the terminology for the entire document. The term "deforestation" as used in the document does not fully correspond to the widely regognized FRA definition of deforestation which is a change of land use from forest to another land use. Deforestation as used in this document is the net loss of forest area which includes the combined effect of deforestation and forest expansion.

Also, the term Forest land as used here correspond to the Forest area (Forest with the FRA definition) at the end of each period and not necessary exactly to forest land or forest land remaining forest land according to IPCC.

I think a clarification of these two concepts in the beginning would help the further

understanding of the document.

---

## Short Comment (SC2) · 14 Oct 2020

When you look at the forest area change, for a certain period, even if it is a negative net change, it can have two components (deforestation, i.e. change of forest to other land use; and forest expansion i.e. change of other land use to forest. The difference is the net effect of these two change processes, and it is not entirely correct to call the net loss deforestation.

Likewise, countries that have a positive net change, still can have deforestation but it is outweighed by the forest expansion.

Further, when area data are not available disaggregated by planted and natural forest, is then the net flux assumed to be due to "deforestation" when total flux is negative and

when positive assigned to forest land remaining forest land?

---

## Short Comment (SC3) · 14 Oct 2020

In section 3 Results reference is made to that results are presented for all UN member states, however the FRA dataset which has been used is covering more countries and territories compared with the number of UN member states.

---

## Referee Comment (RC1) · Marieke Sandker (Referee) · 22 Oct 2020

Overall assessment The topic of this paper is highly relevant and timely. The paper is well written and the comparison with GHG inventory reporting under the UNFCCC is useful as well as the reference in the discussion to the difference between anthropogenic and natural fluxes. This work is trying to fill an important data gap with high uncertainties. The methodology section would benefit from a more detailed description on how the data was obtained and how missing data was treated. Finally, some concerns exist concerning the assessment of removals/emissions in remaining forest land. This may merit further elaboration in the discussion perhaps also comparing with studies assessing natural fluxes (i.e. all fluxes), in addition to the comparison with only anthropogenic fluxes in the GHG inventories.

[Figure]

Overall comments Equation 1 suggests carbon stock changes in forest land remaining forest land are calculated comparing country reported carbon stocks for different years. Are national correspondents aware that the estimates provided here would be used to assess emissions/removals from forest land remaining forest land? Would it not be more straightforward to ask countries directly whether they can report on the change of their country's carbon stocks in forest land remaining forest land (instead of asking them to report carbon stock in different points in time)? For example: if a country has 100,000 ha forest in the year 2000 of which 90,000ha is primary forest and 10,000ha is secondary forest, and in 2010 that country has 95,000 ha of forest if which 90,000 primary forest and 5,000ha secondary forest, the average carbon stock in the forest in time 2 has gone up but it could be incorrect to interpret this as removals in forest land remaining forest land? Could this be a concern?

In addition, many countries will not have multiple assessments and therefore report carbon stock only for one year without this meaning the flux in forest land remaining forest land being zero – on the other hand countries may have estimates for different years using different methodologies making them not directly comparable. Looking into the spreadsheet, DRC's forest land remaining forest land has zero emissions/removals. This seems highly unlikely? DRC's GHG inventory seems to suggest F>F to remove approx. 50 million tCO2eq/yr? How does the study deal with these limitations?

Pan et al 2011 suggests forests globally to on average remove 4.0 bln tCO2eq, this study suggests net emissions of 0.4 bln tCO2eq - that's a 4.4 billion tCO2 gap. Could it be that the above mentioned issues (countries lacking data on evolution of carbon stock + interpretation of stable reported values as zero fluxes) contribute to the explanation of this gap?

What is the data input for deforestation area estimates? Are you using country reported deforestation numbers gap-filled with negative net forest area changes in case countries did not provide deforestation estimates? If so, it would be good to describe this in the methodology. If not, why is this not used?

It seems removals from non-forest land converted into forest land are not estimated. Why not? I have not seen it explained in the article – it would be good to highlight this such that the reader is made aware this flux is missing?

Detailed comments Line 43/44: Estimates of CO2 emissions and removals from forest land were computed following the carbon stock change method of the 2006 IPCC (2006) guidelines, Vol. 2 and 3, at Tier 3, approach 1 (Federici et al., 2015; FAO, 2020a). ïČŸ Is this an accurate description of the methodology applied? According to Jim Penman, the stock change method requires repeated field measurements – very few countries would have this. Any form of AD x EF is considered gain-loss https://www.reddcompass.org/uncertainty?uri=_Toc372288937.html%23_Toc372288937&ver=v1#gfoi-mgd-content Wouldn't the approach rather be a mix of stock change and gain-loss? ïČŸ Wouldn't the assessment be better described as Tier 1 level, or at best Tier 2 since country estimates are used, rather than Tier 3? Perhaps a model was applied here but Tier 3 does also imply an increase in accuracy so perhaps a different type of model is implied for Tier 3. Line 60: it would be helpful if the paper would explain what is meant with net deforestation (net area change? excluding temporary tree cover loss? considering the carbon contents in the replacing landuse?) Line 88: Should this be forest area flux? Line 235: First, the good agreement between the FAO estimates and country reports implies that the definition of forest land use underlying both FAO and UNFCCC reporting was consistent, i.e., all managed and hence the emissions were considered all anthropogenic. ïČŸ Can this truly be concluded? As aggregate values the reported emissions/removals were less than 15% different but what were the differences at country level, were these not much larger? Even if they were comparable I still wouldn't conclude from it they are both managed. In theory with the FRA forest area you calculate using the full forest extent without making a distinction between managed and unmanaged so why conclude this? Editorial: Line 51: as show in more detail below > as shown in more detail below Line 77: . . .on forest land proper, and from deforestation. ïČŸ I don't understand this Line 90/91: over or underestimates > over or underestimated Line 93/94: (which is the primary variable measured, from

which total carbon stock is obtained by are multiplication) ïČŸ I don't understand this
Line 225: For Indonesia, the new FAO estimates (as well as those based on the FRA
2015) had greatly overestimated country reported data for 1991-2000 ïČŸ Do you
mean to say FAO estimates were much higher than GHG inventory reported data?
Harmonize the 1991-2020/1990-2020 period annotation

---

## Short Comment (SC5) · 29 Oct 2020

Why were countries divided/categorised into either Annex I or non-Annex I? This categorization denotes the KP. And if it is KP data, then the US is not included? And since 2012, 1-2 other developed countries have stepped out. It may be more helpful to categorize by developed and developing countries to include all Parties to the UNFCCC.

In terms of the figures, for developing countries why has Brazil and Indonesia (despite them being huge forest countries) being highlighted. But according to the data provided in the article, the growing problem lies in Africa. Is it possible to present what is happening in the DRC, if you are only selecting large forest countries.

[Figure]

Also, in these figures, you mention "deforestation" but in the y-axis, it refers to MtCO2 as emissions.

It is not clear why the 2 categories, naturally regenerating and planted forests were chosen, but how did you compare these to the UNFCCC reported categories. The selected categories would apply to the KP, article 3.3 and 3.4. But developing countries do not report on these Articles. Under the Convention, countries report on forest lands remaining forest lands (could be natural forests) or forest lands converted to other land uses or other land uses converted to forests (planted forest), following IPCC LU categorization.
* * *

---

## Short Comment (SC6) · 29 Oct 2020

Line 109 "were summarized by UNFCCC annex," What is this "UNFCCC annex"?

Line 155 "with very large uncertainty (about 70% as discussed)," As discussed where?

Line 130 Asia was the third region in terms of deforestation emissions, with decreasing trends since 2010, i.e., from 0.6 Gt CO2 yr-1 (2011-2015) to 0.4 Gt CO2 yr-1 (2016-2020) "Since 2010" would mean that there was also a decreasing trend in the period 2011-2015, and with further decrease in 2016-2020. So what were the emissions in 2005-2010, that could indicate this decreasing trend "since 2010"?

Actually the use of the different periods to compare and show trends is rather confusing. At one point, you are comparing 2011-2015 to 2016-2020, then at another point,

you are comparing 1991-2010 or using a timeframe of 1990-2020. If the purpose is to compare or show what has changed between FRA 2015 and FRA 2020, this is rather confusing.

Line 152 Answers what I noted above, but I had to read all the way down before I come to this as one of the purposes of this paper: By combining information on the two periods 2011-2015 and 2016-2020 to obtain for the first time a picture of the 2011-2020 decade, it follows that in 2011-2020 the net contribution of forests to the atmosphere were virtually zero (less than 200 Mt CO2 yr-1),

But the above still doesn't make clear to me the different time frames compared as noted above.

Line 224 For Indonesia, the new FAO estimates (as well as those based on the FRA 2015) had greatly overestimated country reported data for 1991-2000, i.e., by factors of over 10.

I understand you are comparing estimates reported to the FAO with estimates derived from the GHG inventory to the BUR? Aren't FAO estimates also country reported" So it is unclear in the 2nd line when you refer to country reported data (to whom, to where?)

---

## Short Comment (SC7) · 29 Oct 2020

During the period 1991-2020, natural forests lost annually more than 2% of their biomass C stocks, corresponding to a mitigation potential of roughly 3 Gt CO2 yr-1 when future C stock losses from deforestation are avoided.

Let me see if I understood this final sentence correctly? You are saying that for the period 1991-2020, natural forests (what the difference between this with the naturally regenerating forest category that you began the article with?) lost 2% of their biomass stocks. And this is equivalent to about 3 Gt CO2/yr loss for this period? And if the biomass stocks were not lost in the first place, this 3 Gt could have been the mitigation potential. Then the sentence ends by saying: "when future C stock losses from D are

avoided" Thus it is not clear how the ending phrase tie in to the first part of the sentence which talks about losses in the period 1991-2020. And what is the impact that you want to show in this final sentence? I think the article ought to conclude with findings that leave an impact, e.g. forest continuing to be a net sink (albeit small) and thus has a role in mitigating climate change.

---

## Short Comment (SC8) · 30 Nov 2020

General: This is a timely and extremely useful paper. It provides a critical and significant update to pervious FAO data. It is critical both to the scientific community who rely heavily on FAO data, and also the policy community in negotiating the Paris Agreement work plan, raising ambition and particularly ahead of the Global Stocktake, as well as in the international review process for inventory data. FAO data forms a key part of evidence for assessment in IPCC reports including the upcoming AR6. The paper is of high quality and high relevance. I strongly urge to publish it as soon as possible. The comments I have are easily dealt with as part of minor revisions.

Specific comments:

[Figure]

Definitions: Some definitions would be helpful earlier in the paper. The paper talks about fluxes due to deforestation (loss of forest area) – Please clarify if this loss of just primary forest area or is it also loss of secondary (regenerated and planted) forest area), I think the latter. Also please clarify that the "forest land" flux includes both increase in forest area due to regeneration and planting, as well as any forest management/degradation or environmental drivers ($CO_2$ fertilisation, climate change) that in net leads to increasing (or decreasing as in Africa) stock in extant forests. Finally, in the discussion (page 6 lines 170-180) you make it clear that the changing stock in "forest lands" should by definition apply to managed land only, but that some countries also report on unmanaged lands, so this is a mix of anthropogenic and non-anthropogenic leading to an overestimated. This info is helpful to include up front as well as in the limitations section.

Section 2.2 limitations and uncertainty: could you include a comment about legacy effects of deforestation prior to 1990 and move the discussion or some comment on unmanaged lands to here.

Comments by line:

Pg 1 line 23 "Remarkably, the new data also suggest an overall net sink of about -0.7 Gt $CO_2$ yr-1 during 2011-2015, never reported before." Do you mean specifically by FAO as the inventories reported a small sink in this period as included in IPCC SRCCL.

Pg 2 line 34 should either refer to the whole IPCC SRCCL, or to chapter 2 Jia et al 2019 where emission estimates are discussed in detail either in addition to or instead of Arneth et al. .

Pg 2 line 34-36, are these umbers the mean across the three different estimates? If so state this. I think it would be better to give the range, or better still the individual numbers as they differ from each other for different reasons. (may also have the opposite trend?)

Pg 2 line 51 "showN"

Pg 3 lines 80-84. Its confusing as you say on include on the fie first two of six categories, then list all six, with 1 being AG and BG biomass and 2 being dead wood. But on line 84 you say "we including only ..living biomass". Please clarify better by listing in the first place what was included, and then what was not included.

Pg 3 line 89-90, you say "two sub-components" but list only one

Pg 5 line 134. I don't really understand this "Results show that remaining forest land (i.e., net of deforestation)" do you mean "not including deforestation." See point above re, helpful to define what you include in "forest land" in increase in area of planted and regenerating forests? As well as change in carbon stock on extant secondary (regrowing or replanted) forest areas. Also double checking whether or not it is including change in carbon stock on primary forests, may be pertinent to Africa comment below.

Pg 5 line 144-148. Again as per definition comments above, to double check this is all loss for carbon stock in extant forests, not loss of forest area of secondary forests which would count as deforestation? And is it in all forests, primary and secondary?

Pg 8 line 240. This whole section is lacking a comparison with global models, while you don't need to go into this in details, I think it is worth highlighting in relation to your concluding sentence. While the NGHGIs fine a small net sink, the global models do not (Friedlingstein et al., 2019, subm; SRCCL, Grassi et al 2018) . Then you can also refer to the findings of Grassi et al that this is mostly because in the modelled definition a lot of the sink in extant forests due to increase in carbon stock is considered to be due to the natural response of forests to environmental change, and is not considered to be anthropogenic in the models. I think its important for both science and policy communities to understand this, especially those more familiar with the results of the global carbon project. It only needs to be a short explanatory comment.

Pg9 line 251. I don't think you can say "never previously detected with this magnitude"

because in your figure the NGHGIs show a similar magnitude. Can you qualify that its never detected at this magnitude by FAO. Also to note that the global DGVM models do detect a large sink in extant forests in both managed and unmanaged lands, but it is not reported as part of anthropogenic flux. This goes back to the point above. You make it sounds like FAO has discovered a large sink no-one previously knew about, when the inventories and models both report this sink.

———————————————

---

## Referee Comment (RC2) · Anonymous Referee #2 · 9 Dec 2020

This paper is a timely important piece of work and data could be useful for upcoming verification of the performance of the REDD+ activities in developing countries.

Main problems at this stage are more on consistencies in presenting the results. 1. Unit of carbon: Authors used Gt CO2, Gg CO2, converting from Mt CO2 to Gg CO2. Although Gt CO2 and Gg CO2 are the same in terms of value, I think authors should use only one type i.e. Gg CO2 because the equations were expressed in terms of Gg CO2, not GtCO2

2. Living biomass: Authors mentioned living biomass to refer to two carbon pools, but they did not refer specifically to aboveground and belowground. It might be useful to say specifically as per the IPCC Guidelines.

[Figure]

3. Five vs Six carbon pools: I believed that IPCC Guidelines refer to five carbon for national reporting. The HWP pool is optional and not eligible for performance verification. Please check this again carefully

4. In Introduction part, authors may want to describe the need for understanding the past carbon emissions under the REDD+ scheme to make this study more relevant to the on-going international policy.

5. Forest flux, forest change (L60): I think these terms are still confusing. Can we add "forest carbon fluxes", "forest area change", etc.

6. (L65) Bi should be TgC (not MtC. TgC is correspoding to Gg while Mt is corresponding to Gt). Please add living biomass (aboveground and belowground)

7. L65, 44/12 ... Mt C should be Tg C (Teragram = 1000000 MgC or GgC/1000)

8. Uncertainties I think this section needs several more references to support the arguments. Please see early comment about six carbon pools

9. Results To be in line with the on-going REDD+ scheme, I think authors should describe the results by simply say Firstly, report the emissions: refer to emissions from deforestation only (please check FREL of the REDD+ Rules) Secondly, report the removals Thirdly, report the net emissions (please check FRL) Although REDD+ is more on developing countries (presumably all Non-Annex 1 countries), it would useful to describe the results in line with the REDD+ Rules or the Warsaw Framework

Authors should also describe, something like ... Our results will be presented by Annex 1 and Non-Annex 1 countries before referring audiences to the Table 1, in which nothing was described earlies.

3.1. can it be Forest Carbon Flux? forest flux could mean many things

L135: i.e. net of deforestation .... I think it should be i.e. after deducting the net deforestation.

From here, authors described the emissions in terms of Gt CO2 but it was Gg CO2 in the equations.

Conclusion From the tone of this writing, this paper seems to be the work of the FAO, not the authors themselves. Please re-think and rewrite if possible.

Table 1: Title here is confusing to me. can it be Estimates of total forest carbon fluxes from deforestation and planted forest by Annex 1 and Non-Annex 1 countries between 1990-2020

---

## Author Comment (AC1) · 2 Mar 2021

Thank you for these important clarifications on definition of deforestation. The revised manuscript has a greatly revised and expanded methods section when these issues are addressed for improved clarity and interpretation of results. Below please find a zip file with replies to all reviewers and comments directly within the original pdf provided by the Journal. We trust that the revised manuscript sufficiently addresses all comments, for which we thank the various authors. In general, the revised manuscript ahs a revised and greatly expanded methods section addressing a variety of issues brought forward by reviewers, from land use definitions, to improved description of equations, expanded discussion on limitations and uncertainties. The manuscript also provides a clearer context within which this effort contributes, namely the assessment

of forest emissions and removals, contributing to the ongoing debate on the differences between terrestrial carbon cycle models and national inventories, but with a main focus on simple use of robust and ready available data.

Please also note the supplement to this comment:
https://essd.copernicus.org/preprints/essd-2020-203/essd-2020-203-AC1-supplement.zip

―――――――――――――――――

---

## Author Response (AR2)

Comments to the Author:
Very positive and thoughtful changes to so many comments. Thank you for using ESSD!

FNT: Thank you

Small additional suggestions and comments (these can occur during proofreading):

Page 1 line 30 and elsewhere: do you want to include the URL for the FAO Emissions database? I know how to find it - perhaps from prior use - but 'new' users/readers will appreciate a direct link? Particularly with so many accessible FAOSTAT options. Links in references all lead to .pdf but here you mean actual FAO or FAOSTAT landing page?

FNT: we have specified a new reference, FAO 2021, which points to the FAOSTAT database

Page 2 line 13 - generated?

FNT: replaced with ''linked to''

Page 3 line 23 - 'uncertainty' is incomplete? Or, uncertainty results in incomplete data? Not sure whether you mean here that country reports lack uncertainty estimates or that inconsistent or incomplete country reporting results in residual uncertainty?

FNT: sentence fixed for clarity. We meant that country reports lack uncertainty estimates, which makes our job more difficult, often having to rely on IPCC generic guidance on this.

Page 6 line 30 - NAI here means non-Annex I? You have not used or defined this term elsewhere.

FNT: now spelled out

Page 7 line 9 - You do not use this unit - Gg - anywhere else? I think you mean Gt?

FNT: we had replaced Gg with kt everywhere else – fixed in this case as well

Page 10 line 32 – forests to change to forests to, proofreaders will pick this up but you might as well make the change now.

FNT: thanks, done.

Page 11 lines 26, 27 - redundancies here? To make access easier, give the DOI in the one-click format https://doi.org/10.5281/zenodo.3941973?

FNT ok

Page 13 References - very strange no references alphabetically before 'FAO'. Earlier version had Arneth et al? Please check. If valid, highly unusual.

FNT: the Arneth et al. has been changed (by IPCC) to IPCC, 2019. It is what it is I guess.